# Analysis of Deep-Learning Methods in an ISO/TS 15066–Compliant Human–Robot Safety Framework

**DOI:** 10.3390/s25237136

**Published:** 2025-11-22

**Authors:** David Bricher, Andreas Müller

**Affiliations:** Institute of Robotics, Johannes Kepler University, 4040 Linz, Austria; a.mueller@jku.at

**Keywords:** human–robot collaboration, human body recognition, human pose estimation, human body part segmentation, human body segmentation, safety-relevant human–robot interaction, ISO/TS 15066, applicable artificial intelligence, deep learning

## Abstract

Over the last years collaborative robots have gained great success in manufacturing applications where human and robot work together in close proximity. However, current ISO/TS-15066-compliant implementations often limit the efficiency of collaborative tasks due to conservative speed restrictions. For this reason, this paper introduces a deep-learning-based human–robot–safety framework (HRSF) that aims at a dynamical adaptation of robot velocities depending on the separation distance between human and robot while respecting maximum biomechanical force and pressure limits. The applicability of the framework was investigated for four different deep learning approaches that can be used for human body extraction: human body recognition, human body segmentation, human pose estimation, and human body part segmentation. Unlike conventional industrial safety systems, the proposed HRSF differentiates individual human body parts from other objects, enabling optimized robot process execution. Experiments demonstrated a quantitative reduction in cycle time of up to 15% compared to conventional safety technology.

## 1. Introduction

The steady reduction of process execution time and improved machine flexibility is a primary goal in industrial automation. In this context, the introduction of collaborative robots (cobots) has emerged as a promising new technology that could particularly handle challenges in manufacturing industries [1,2]. Since the design of cobots allows operation in close proximity to humans, it is mandatory to guarantee human safety under all circumstances, i.e., potential collisions between human and robot shall not cause physical harm. For this reason, the main requirements for the integration of collaborative systems in industrial environments have been laid down in the technical specification of the International Organization for Standardization ISO/TS 15066 [3].

In practice, safety assessments typically determine the expected force and pressure levels for given robot poses and speeds before plant installation. To ensure compliance, robot velocities are often limited to the lowest admissible value across the workspace, which can significantly constrain process efficiency, particularly in high-throughput tasks. Consequently, even in situations where a collision is unlikely, the process flow may be unnecessarily delayed.

Existing industrial solutions, such as laser scanners or proximity sensors, provide conservative separation monitoring but lack the capability to differentiate humans from other moving objects or to identify individual body parts—both critical for dynamic safety regulation under ISO/TS 15066. Vision-based methods using RGB-D data have been explored [4,5,6], with depth information enabling moving-object detection [7]. However, traditional approaches generally treat all detected objects uniformly, failing to exploit the potential of selectively optimizing robot motion based on human-specific spatial information. To address these limitations, we propose a conceptual human–robot–safety framework (HRSF) that integrates deep learning with RGB-D input to enable accurate spatial localization of both the human body and individual body parts. The framework dynamically adjusts robot velocity according to the required separation distance for each body part while maximizing process efficiency. The HRSF is a performance and feasibility study for vision-based separation monitoring (SSM) under ISO/TS 15066 constraints. While it demonstrates how body-part-aware perception could influence task execution times and motion efficiency, it is not a certified or deployable safety system, and the experiments are intended to evaluate performance and conceptual adherence rather than formal compliance.This approach explicitly leverages the predictive capabilities of deep learning to distinguish humans from other objects and to account for variable poses, providing an advantage over conventional sensor-based methods. The primary research question guiding this study was as follows: Can the proposed deep-learning-based HRSF improve process execution times under constraints of ISO/TS 15066 compared to state-of-the-art industrial safety technology? To investigate this, we systematically evaluated the accuracy and robustness of multiple deep learning architectures in a collaborative manufacturing scenario.

To clearly articulate the novelty of our approach, we summarize the main contributions of this work:**A body-part-aware RGB-D safety framework aligned with ISO/TS 15066.** We introduce a human–robot–safety framework (HRSF) that uses deep learning to estimate the 3D locations of individual human body parts and maps them to the body-part-specific safety limits specified in ISO/TS 15066. In contrast to existing whole-body detection approaches, the framework enables differentiated separation distances that reflect the varying biomechanical tolerances of different human regions.**A systematic comparison of deep learning architectures for safety-critical spatial perception.** We benchmark multiple state-of-the-art RGB-D models for body and body part localization with respect to accuracy, robustness, latency, and failure modes—metrics that have rarely been evaluated together in prior vision-based human–robot collaboration (HRC) safety work. This analysis supports a more realistic assessment of whether body part granularity can meaningfully improve safety-aware robot motion.**A dynamic velocity-adaptation scheme based on body part proximity.** We implement and evaluate a separation-monitoring controller that adjusts robot velocity according to the nearest detected body part and its corresponding ISO/TS 15066 threshold. This differentiates our work from existing RGB-D safety systems that rely on uniform safety margins and cannot exploit less conservative limits when nonsensitive body regions are closest.**Experimental validation in a real collaborative manufacturing scenario.** Using a KUKA LBR iiwa 7-DOF robot (KUKA AG, Augsburg, Germany) we demonstrate the feasibility of the proposed framework in a representative screwing task and measured the operational impact of body-part-aware velocity regulation. Although limited in subject number, task diversity, and repetitions, these experiments provide initial evidence for how fine-grained human perception can influence cycle time under real processing latencies.

Following this, we provide a comparative overview of the related approaches, including conventional industrial sensing, RGB-D and skeleton-based human detection, and other learned segmentation methods. Table 1 contrasts each method with our HRSF in terms of sensing modality, alignment with ISO/TS 15066, body part awareness, dynamic velocity adaptation capability, and experimental validation. This comparison highlights where the proposed framework advances the state of the art and clarifies the specific gaps it addresses in vision-based HRC safety.

## 2. Safety Aspects According to ISO/TS 15066

In the manufacturing industry the use of cobots has allowed man and machine to work in shared work places. Especially for those tasks where human and robot work together in close proximity, the robot speed must be reduced or the robot must be stopped in order to avoid safety-critical collision scenarios. Typically, the robot speed is adjusted to a constant value that is in accordance with the most restrictive collision situation that is possible at a particular work place, which ultimately increases the robot cycle time for process execution. Indeed, when all body parts of humans within the collaborative workspace are located at a predefined safety distance from the machine, the robot speed might be increased in order to reduce the robot cycle time. To this end, the proposed HRSF applies different deep learning approaches for the spatial localization of humans in the shared workspace, and accordingly, the robot speed is adapted to the maximal value allowed. In order to apply the HRSF in manufacturing environments, it must comply with the core requirements from ISO/TS 15066.

In general, the ISO/TS 15066 regulation distinguishes four different types of collaborative operation:Safety-rated monitored stop.Hand guiding.Speed and separation monitoring (SSM).Power and force limiting (PFL).

Most research activities have mainly focused on either one of these four collaboration modes. For speed and separation monitoring (SSM), a common field of research aims to avoid collisions with real-time path planning methods [19,20], 3D dynamic safety zones [21], approaches for the anticipation of human motion with gradient optimization techniques [22], stochastic trajectory optimization [23], optimization-based planning in dynamic environments [24], or with learning-based methods [25], whereas for power and force limiting (PFL), the effects of collisions are of particular interest [26,27,28,29,30].

In contrast to the above cited research, the proposed HRSF combines the normative guidelines of collaboration operation modes (c) and (d). At first, the regulations for SSM are applied when operators are identified in the shared workspace. Accordingly, whenever a critical distance is reached, the robot speed is reduced to a value that corresponds to the requirements for an operation in PFL mode. Consequently, the framework does not exclude collisions per se but reduces the robot velocity according to the necessary separation distance between human and robot to a maximum level that does not violate the prescribed biomechanical force and pressure thresholds. This allows for the operation of the robot in the most efficient way, i.e., with the maximum velocity allowed that will not endanger humans. The proposed framework can be used for any type of industrial robot system. However, since industrial heavy payload robots are mainly equipped without power- and force-limiting sensors, close proximity of human and machine in shared workspaces can only be exploited if a framework combining SSM and PFL and is used. To this end, this study solely focused on the framework usage with collaborative robots.

## 3. Localization of Humans and Human Body Parts in the Workspace

### 3.1. Relevant Deep Learning Approaches

To safely adapt robot velocities, humans entering the shared workspace must be detected reliably under varying illumination, viewpoints, and body postures. Modern deep learning methods clearly outperform classical computer-vision techniques for this task [31]. Depending on the particular recognition task, different deep learning approaches can be applied. In the following, two classes of algorithms are distinguished, which are either human body related or human body part related.

#### 3.1.1. Human Body Recognition

Current state-of-the-art image-based human detectors rely on convolutional neural networks (CNNs). These methods typically output a bounding box—defined by its center ($x_{C}$, $y_{C}$), width *w*, and height *h*—along with a classification score. Two main architectural families dominate current research:Region-proposal networks such as R-CNN and Faster R-CNN [32,33,34,35].Regression-based detectors such as MultiBox, SSD, or YOLOv4 [36,37,38], which jointly perform localization and classification.

#### 3.1.2. Human Body Segmentation

In applications where pixel-level classification is needed, segmentation approaches provide richer spatial information than do bounding boxes. CNN-based models represent the current benchmark for segmentation tasks [39,40,41]. Mask R-CNN [42] remains one of the most accurate frameworks, using an ROI-Align module to reduce feature misalignment and an additional mask-prediction branch. Although faster alternatives such as YOLACT++ [43] exist, Mask R-CNN generally achieves higher segmentation accuracy.

#### 3.1.3. Human Pose Estimation

Human pose estimation aims to determine the positions of anatomical key points. Recent CNN-based methods outperform earlier model-based and decision-tree approaches [44]. Early regression-based methods [45,46,47,48] predicted joint coordinates directly from images. Later work demonstrated that predicting confidence (belief) maps for each joint significantly improves accuracy [49,50], as these maps capture spatial dependencies more effectively.

#### 3.1.4. Human Body Part Segmentation

Human body part segmentation extends semantic segmentation by distinguishing between individual body regions. Due to limited labeled datasets, modern approaches such as [51] augment training data through synthetic generation and pose-guided refinement. These methods leverage key-point alignments, morphing, and dedicated refinement networks to improve per-part accuracy.

### 3.2. Selected Deep Learning Approaches

Depending on the safety function, it may be necessary to identify either the closest human body point or the closest specific body part. Therefore, the framework evaluates both body-level and part-level deep learning models. Human detection, segmentation, and pose estimation methods are trained on MS COCO [52], while body part segmentation models use the PASCAL-Part dataset [53]. All selected architectures used within the framework are summarized in Table 2.

### 3.3. Extraction of Depth Information

Within the HRSF the extraction of spatial human body information is carried out with RGB-D data, captured from an Intel Realsense D435i camera (Intel Corporation, Santa Clara, CA, USA). While the surveyed deep learning methods are applied to RGB input images, the gathered depth information is aligned with the color image data in order to bring the human body information into a spatial context. According to the applied type of algorithm, the extraction of depth information distinguishes for two different classes:ADetermination of the minimal separation distance of the closest body point to a hazardous area.BDetermination of the separation distance for individual body parts.

Since the extracted spatial body information is generally related to the camera coordinate frame $F_{Cam}$, extrinsic parameters are applied in order to determine minimal separation distances with regard to the robot world coordinate frame $F_{w}$.

#### 3.3.1. Minimal Separation Distance for a Single Body Point

For human body recognition and human body segmentation, the desired minimal separation distance is obtained by determining the body point within the bounding box or the human surrounding contour closest to the robot world coordinate origin (Figure 1).

Due to stereo mismatching and aliasing artifacts, a determination of depth information might lead to the occurrence of small depth values close to zero which do not conform with reality. Thus, a lower threshold level $d_{thres}$ is introduced, and all depth data with $d_{min}$ < $d_{thres}$ are neglected.

#### 3.3.2. Separation Distance for Individual Body Parts

The separation distances determined by the recognition of individual human body parts are mainly influenced by the accuracy and robustness of the spatial body part predictions.

For human pose estimation, the body joint predictions correspond to specific coordinates in the image plane. Thus, it would be possible to assign a single depth value to each body joint coordinate. Due to stereo mismatching and occlusion, a determination of depth information on the pixel level might lead to erroneous depth values, which do not conform with reality. Therefore, again all depth data points with $d_{min}$ < $d_{thres}$ are rejected, and instead, the mean depth value $d_{mean}$ within a prescribed region of interest $A_{ROI}$ (e.g., small windows: 10 × 10 pixels) is determined (for comparisons, see Figure 2a).

In contrast to that in human pose estimation, the depth information in human body part segmentation can be gathered for all image points that are assigned to a specific body part; i.e., for each image point which is associated to a specific body part, the corresponding depth value is extracted. From all of these body-part-related depth values, the minimal spatial separation distance is now determined for each body part individually. Thereby, a more robust spatial estimation of the corresponding body parts can be achieved compared to human pose estimation. An illustration of the depth extraction for human body part segmentation is given in Figure 2b.

## 4. Determination of ISO-Relevant Safety Parameters for Specific Robotic Systems

The safety-relevant parameters for SSM and PFL are specific to a particular implementation of the proposed HRSF. For SSM the corresponding parameter is the minimum separation distance $S_{p}$ at which the robot velocity can still be reduced to the corresponding safety parameters for PFL, which are the body-part-specific maximum robot velocities ${\dot{z}}_{max}$ that comply with the biomechanical force and pressure thresholds of ISO/TS 15066. In addition to ISO/TS 15066, the determination and validation of these parameters must also consider the functional safety requirements defined in ISO 13849-1/-2 [55] and IEC 61508 [56], which prescribe performance levels (PLs) or safety integrity levels (SILs) for safety-related control functions. These standards provide guidance on evaluating hardware reliability, diagnostic coverage, and common-cause failures, all of which directly influence the acceptable reaction times and safety margins for SSM and PFL implementations. Since the parameters highly depend on the hardware (e.g., robot) and software (e.g., algorithm) components used, the following section describes generic methods for their determination. For the sake of comparison, the proposed approaches are applied to the different algorithms, as discussed in Section 3.

### 4.1. Minimum Separation Distance $S_{p}$

The guidelines for SSM in ISO/TS 15066 define the minimum separation distance required between human and robot $S_{p}$ before adapting the robot velocity as(1)$$\begin{array}{c} S_{p}=S_{h}+S_{r}+S_{s}+C+Z_{d}+Z_{r} \end{array}$$
with $S_{h}$ and $S_{r}$ being the distance contributions attributable to the reaction time for sensing the current human and robot location, *C* describing the intrusion distance in the perceptive field, and $Z_{d}$ and $Z_{r}$ respectively being distance contributions corresponding to the uncertainties in human and robot position sensing. The distance contribution $S_{s}$ corresponding to the robot system’s stopping is neglected within the framework since it is the main aim of the framework to avoid robot-stopping behavior.

From a functional safety perspective, both ISO 13849-1 and IEC 61508 emphasize that the determination of reaction times and corresponding distance contributions must incorporate validated safety-related software execution times, sensor update rates, and fault-tolerant processing intervals. In practice, this means that the measured values for $S_{h}$ and $S_{r}$ and the uncertainty terms $Z_{d}$ and $Z_{r}$ must include worst-case execution times and consider hardware fault behavior to achieve the required PL or SIL. These standards also mandate verification and validation procedures ensuring that risk-reduction measures—such as velocity reduction and minimum distance enforcement—are implemented with adequate reliability.

In the following, the individual contributions to $S_{p}$ are analyzed in more detail by means of experimental tests.

#### 4.1.1. Distance Due to Human Motion $S_{h}$

The contribution in the separation distance that corresponds to human motion is given in ISO/TS 15066 as(2)$$\begin{array}{c} S_{h}=\int_{t_{0}}^{t_{0}+t_{r}}v_{h}\;\cdot \;dt. \end{array}$$

For all cases where the human speed cannot be monitored with specific sensor systems, a constant velocity $v_{h}$ corresponding to 1.6 m/s for separation distances above 0.5 m and 2.0 m/s for distances below 0.5 m is assumed, respectively. Within the framework, no additional sensors are attached to the human body, which means that $S_{h}$ can be characterized by the reaction time $t_{r}$ of the robot, i.e., the latency. In the context of the framework, the latency is defined as the required time for the sensing system to perceive the human body up to the moment when the robot has decreased its velocity to a safety-conforming level.

In addition to the definition provided in ISO/TS 15066, both ISO 13849-1 and IEC 61508 have direct implications for determining $S_{h}$. ISO 13849 requires the reaction time of the safety-related control system to be evaluated with respect to its achieved performance level (PL), meaning that the latency used in the separation-distance calculation must incorporate the worst-case response time of all components contributing to the safety function. Similarly, IEC 61508 stipulates that the execution times of safety-related software, diagnostic functions, and signal-processing chains must be validated under worst-case conditions when determining the safety integrity level (SIL). Therefore, the experimentally measured latency $t_{Lat-Max}$ must include conservative margins covering hardware fault behavior, communication jitter between computation modules, and the maximum expected cycle time of the sensing and control tasks. These requirements ensure that the resulting $S_{h}$ complies not only with biomechanical safety constraints but also with the probabilistic and systematic reliability constraints of functional safety standards. Since the HRSF and the robot path planning are running on individual computation modules and in order not to introduce further latencies, a simple visual method is used to synchronize them.

The latency measurements are triggered by the flashing of a light bulb that is initiated by the robot controller. Within the human sensing node, at first the illumination of the light bulb is checked before human body information is extracted from the RGB-D input data. Each time a light bulb flash is registered, the robot velocity is reduced to the minimum level allowed. The maximum latency levels are observed when the light bulb flash is initiated directly after an RGB-image has been captured; i.e., in this case most processing steps are carried out twice. After the robot has reached the desired velocity limit, the latency measurement is stopped. For each of the algorithms analyzed, the latency measurements are carried out for 300 s in order to determine a relative estimate of the maximum latency levels $t_{Lat-Max}$ occurring. Ultimately, the latency is determined by the following factors: the image-capturing time $t_{Cap}$, the inference time $t_{Alg}$ of the algorithm, the spatial information extraction time $t_{3D}$. and the robot velocity adjustment time $t_{Adj}$. An overview of the obtained algorithm-specific latencies is given in Figure 3.

The algorithm-specific maximum separation distances $S_{h}$ that can be derived from $t_{Lat-Max}$ are given in Table 3. A more rigorous description of the applied method as well as of the individual latency contributions are given in [57]. The latencies refer to the use of an NVIDIA Titan RTX GPU (NVIDIA Corporation, Santa Clara, CA, USA) for human body information extraction.

Apart from the human sensing latency, the current robot configuration also influences the minimal separation distance. The proposed framework determines $S_{h}$ with regard to the robot world coordinate frame $F_{w}$. Indeed, any point on the robot surface might collide with the human body. Consequently, the minimum separation distance needs to be determined for all parts of the robot. In order to minimize the computational expense and determine the separation distance between any point of the human body and the robot surface, a cuboid protective hull $P_{q}^{R}$ is used to describe the robot configuration *q* at a current time step. In order to determine the robot protective hull, the robot-specific Denavit–Hartenberg parameters can be used to extract the individual robot link positions from robot forward kinematics. The minimum and maximum robot deflection in each spatial position can be used to describe the protective hull with $P_{q}^{R}$ = ($x_{min}^{R}$, $x_{max}^{R}$, $y_{min}^{R}$, $y_{max}^{R}$, $z_{min}^{R}$, $z_{max}^{R}$). Accordingly, the minimum separation distance is adapted as the closest distance between a body part and the protective hull. For a comparison, please see Figure 4.

#### 4.1.2. Position Prediction Uncertainty $Z_{d}$

In addition to algorithm-specific latency, the spatial position prediction uncertainty $Z_{d}$ constitutes a major factor influencing the required minimal separation distance. To quantify the individual error characteristics of the evaluated deep learning approaches, the spatial predictions produced by each algorithm were compared with ground-truth measurements obtained from a marker-based motion capture system. The deviation of the algorithm predictions were compared with 25 different markers that were attached to a human subject. Prior to data collection, all markers were placed according to the initial estimates provided by the respective models. Aside from a shared head marker, nine distinct marker positions were defined for the human pose estimation and human body part segmentation methods. For the human-body-centric approaches, these markers were also considered, supplemented by four additional torso markers and two markers placed near the elbows to improve the extraction of torso and upper-limb poses. The complete set of marker positions and their assignment to the respective methods are illustrated in Figure 5. Measurements were conducted for multiple static postures and a range of dynamic movements. An overview of all tested motion scenarios is provided in Figure 6. Each measurement was carried out for 3 min, ultimately corresponding to 1000 data samples acquired per measurement and body part.

The performance of the deep learning models was evaluated in terms of accuracy and robustness. Accuracy was assessed independently for each spatial coordinate and expressed as the mean offset and standard deviation relative to the marker-based ground truth. For body-part–centric approaches, each predicted body part position was compared to its corresponding marker, whereas for human-body–centric methods, predictions were compared to the marker closest to the camera. Robustness was measured by the proportion of failed detections, defined as instances in which no prediction was produced (e.g., due to occlusion) or in which the estimated depth value fell outside predefined limits (i.e., depth values smaller than 0.5 m and respectively larger than 8 m). These robustness metrics served as an empirical basis for estimating perception-system reliability parameters relevant to safety-function validation under ISO 13849-1 (e.g., DC and MTTFd considerations) and IEC 61508 (e.g., diagnostic coverage and dangerous failure rates).

A detailed summary of all experimental results is provided in the Appendix A (compare Table A1, Table A2, Table A3, Table A4 and Table A5). Upper error bounds for each spatial direction were computed by summing the mean offsets and standard deviations over all measurement conditions, yielding a general upper error estimate for each method. For body-part–centric approaches, per-body-part error bounds could also be extracted if required. The results show that the error range for human pose estimation is in general higher compared to human body part segmentation. Indeed, human pose estimation also struggles in detecting individual body parts when they are partially occluded. Within several of the analyzed scenarios, comparatively high mispredictions were obtained for arm positions. This behavior was observed for both the standing and sitting positions as well as for dynamic movements. Especially for sitting poses at high distances to the camera, the detection of stretched arms becomes challenging for both body-part-related approaches. An extensive overview of all spatial body part deviations for standing poses is given in Table A3, those for sitting poses in Table A4, and those for dynamic movements in Table A5.

The resulting estimates are presented in Table 4 and clearly demonstrate that the body-part-centric methods outperform the human-body-centric approaches in terms of spatial prediction accuracy. From the results obtained for human-body-related algorithms, one can conclude that at distances below 4 m, the depth resolution Δz achieves detection accuracies < 10 cm. While similar accuracy levels were observed for Δx, much stronger fluctuations were obtained for Δy. Independent of the method and scenario investigated, maximal error deviations > 30 cm were observed for Δy. This can be mainly explained by the fact that the predicted body points were compared to a limited amount of ground-truth marker data. Consequently, there were different body points that might have shared similar separation distances as the closest marker to the camera but did not necessarily have to coincide with the particular marker position.

In contrast to the analyzed human-body-related approaches, the results obtained for human-body-part-related techniques indicate a more precise prediction behavior. Even though at separation distances below 4 m, the obtained depth tolerance level of 15 cm was slightly worse, compared to the deviations observed for human body related methods, the lateral error range was significantly better by reaching 7 cm for most body parts analyzed. At higher distances discrepancies mainly appear for body extremities, e.g., the right upper arm. Particularly, at distances of 6 m, the depth estimation error exceeded levels of 0.5 m, which in turn suggests that an accurate depth resolution is no longer possible for particular body parts. In contrast, the lateral deviations increased only slightly with the distance to the camera, which indicates that the most limiting factor for an accurate spatial body part prediction is the depth sensor resolution. Among the different static, sitting, and moving measurement scenarios, higher discrepancy levels were observed with increasing distances to the camera. Particularly, at distances of >5 m error, levels above 0.5 m would have to be considered. Overall, the results obtained for human body segmentation show a less fluctuating behavior compared to human body recognition.

On the basis of the discrepancy results obtained, the position prediction error $Z_{d}$ should be estimated individually for each method analyzed. Generally, the depth resolution is a limiting factor at separation distances > 6 m and would thus highly increase the error bound estimation. Due to latency separation distance contributions < 1.4 m and limited robot ranges of approximately 1 m, it appears plausible to restrict the error estimation only to those measurement results that are below a distance of 6 m from the camera origin. By applying this assumption, it is, however, necessary to consider that the camera should be located within a tolerable distance to the world coordinate origin in order to guarantee that the separation distance is determined in accordance with the estimated error bounds; i.e., scenarios with minimal human-robot separation distances below 1.5 m must be avoided when the human–camera distance exceeds 6 m.

For human-body-related approaches, it would be generally possible to use error estimations for each body part individually, but for the proposed framework, a general mean error is determined from the individual upper body part error estimations. An overview of all position prediction error estimations is given in Table 4 and emphasizes that both human body-part-related approaches outperform human-body-related methods in terms of their prediction accuracy. Furthermore, it can be concluded that both segmentation techniques outperform the concurring human-body- and human-body-part-related methods analyzed. An extensive overview of the individual spatial deviations at varying distances to the camera is given in Figure A1, Figure A2 and Figure A3. Overall, the results indicate that human body part segmentation provides the highest accuracy and robustness for spatial body part localization using RGB-D data, achieving depth tolerances below 15 cm and lateral errors below 5 cm for all body parts. Moreover, the findings confirm that the Intel RealSense depth camera is capable of reliably resolving human body part depths at distances up to 4 m. Such validated error bounds and detection-failure characteristics constitute essential input parameters for the design and verification of perception-dependent safety functions in accordance with ISO 13849 and IEC 61508. At certain poses the investigated recognition approach is not able to identify human bodies at all.

#### 4.1.3. Intrusion Distance *C*

The intrusion distance *C* is defined as the distance a body part can permeate the sensing field before being detected. In most cases, the applied industrial safety technologies consist of laser light beam systems. The intrusion distance contribution for these safety system is described in the normative guideline ISO 13855 [58] as(3)$$\begin{array}{c} C=8(d-14) \end{array}$$
with *d* denoting the sensor detection capacity in mm. Since the intrusion distance is defined in ISO 13855 solely for two-dimensional protective field sensors, the contribution of a 3D sensing system can be derived from the depth-dependent sensor resolution. Thus, at distances of 4 m to the applied RGB-D camera, the per-pixel resolution corresponds to 8.5 mm and 6.5 mm in the *x*- and *y*-directions, respectively. The ISO regulation proposes to neglect contributions < 14 mm.

In depth direction, the normative guidelines of ISO 13855 do no long hold since the RGB-D sensor does not monitor the intrusion to the workspace at a fixed sensing height but rather over a continuous range. Consequently, the intrusion distance contribution is mainly characterized by the depth uncertainty that is already included in the position prediction uncertainty $Z_{d}$. Thus, due to the applied RGB-D technology, the contribution is not further considered within the determination of the minimum separation distance.

#### 4.1.4. Robot Latency Contribution $S_{r}$

The robot latency contribution $S_{r}$ to the separation distance arises from the robot joint angle querying process. Due to the processing time required from querying the current robot configuration up to the point in time when the minimum separation distance is determined, the robot can still move on its path with maximum speed. A mean latency of 3 ms can be determined experimentally within the HRSF, which corresponds to an upper error bound of approximately 5 mm at a maximum robot velocity of 1.6 m/s.

#### 4.1.5. Robot Positioning Uncertainty $Z_{r}$

The robot positioning uncertainty $Z_{r}$ can be derived from potential geometric deviations of robot lengths that are used for the determination of the robot protective hull. For the estimation of this contribution, the robot forward kinematics prediction at the tool center point was compared with the manufacturer’s robot model position. In total, mean errors of 8 mm, 7 mm, and 11 mm were determined in the *x*-, *y*-, and *z*-directions. Compared to the obtained positioning error, the robot repetitive accuracy of 0.1 mm for the KUKA iiwa is not significant for further consideration.

### 4.2. Maximum Robot Velocities ${\dot{z}}_{Max}$

The maximal speed of the robot (more precisely of the tool or the part that is actually colliding) is linked with the body-part-dependent biomechanical force and pressure thresholds given in ISO/TS 15066. A generalized description of the relationship of robot velocities and resulting forces and pressures during collisions is not trivial since it depends, for example, on the tools mounted on the robot and on the actual robot pose in relation to the environment; for a comparison, see, e.g., [59]. Moreover, it must be distinguished between transient and quasi-static contact situations. At the moment, the HRSF deals with quasi-static situations. Therefore, in the experiments, the maximum velocity was derived from the more conservative quasi-static biomechanical force and pressure limits.

In this context, two extreme cases are analyzed. At first, the maximum force and pressure levels are determined for collisions where the robot motion is directed towards the operator. In the other case, vertical robot movements are considered that can potentially lead to squeezing of human body parts.

The occurring forces and pressures during collisions were determined using a GTE CBSF-75 Basic force measuring device (GTE Industrieelektronik GmBH, Viersen, Germany). Within the measurement setup, the safety controller was adjusted such that the robot motion stops when the external torque levels exceeded 20 Nm in one of the seven robot joints. Force and pressure measurements were carried out for five different velocities and repeated ten times. An overview of the measured force and pressure levels in dependence of the Cartesian robot velocity is given in Figure 7. The obtained results show that for collision situations where the robot approached toward the operator, the adjusted velocity highly depended on the released force levels, while for vertical movements, the pressure exerted also played a significant role. From the results one can derive the body-part-dependent maximum Cartesian robot speed allowed for this specific task, as listed in Table 5.

## 5. The Human–Robot–Safety Framework

The architecture of the HRSF consists of several building blocks that can be freely exchanged depending on the particular hardware used. An overview of the framework architecture is given in Figure 8.

The robot path planning is executed separately from the human recognition algorithms on an individual building block, denoted in Figure 8 as (1). The maximal robot speed and the maximally allowed joint torques (which are used to limit the interaction forces) are also set in this module. For the employed KUKA iiwa robot, adjustment of the pre-planned path was pursued on the robot controller using specific JAVA libraries.

The current robot position is obtained by the robot location building block (2) via a real-time interface that queries current joint angles *q* from the robot controller. The KUKA iiwa system offers a real-time package—Fast Robot Interface (FRI)–which allows query of robot positions within time intervals of less than 5 ms. A kinematic robot model is used to compute a cuboid-shaped robot protective hull $P_{q}^{R}$ (see Figure 4).

The human body (part) extraction is executed as a separate building block (3), running on a computation node with sufficient processing power, i.e., multiple GPU cores. Within this building block the algorithm-specific body point(s) closest to the robot world coordinate frame $F_{w}$ are determined from the gathered RGB-D information.

After the current robot pose and the most critical human body points are identified, both spatial information are sent via TCP/IP protocol to building block (4), which evaluates the current separation distance between human and robot. Here, the separation distance is determined as the minimal distance of each of the determined body points to all faces of the protective hull. The obtained separation distances are compared with the minimum level allowed depending on the applied algorithm; i.e., the algorithm-specific safety contributions are used in this building block in order to determine if the robot velocity needs to be adjusted. If a particular body part violates the allowed minimum separation distance, the robot velocity is reduced. Since the velocity of the particular KUKA iiwa robot can only be adapted dynamically with “safe” input signals, the information of the maximum robot velocity is first sent to a safety PLC via OPC UA protocol. Accordingly, the obtained signals are transferred to “safe” output signals that are sent to the robot controller. Depending on the obtained information, the robot velocity is adapted.

## 6. Experimental Validation

### 6.1. Test Scenario

The HRSF was tested in a real-world manufacturing scenario that included screwing procedures. Within the experiments carried out, the robot tightened three different screws on an engine block while the worker carried out other workings tasks at the same time. Ultimately, the analyzed use case could be divided into three different phases of interaction of human and robot:Coexistence: Human and robot are located far away from each other but are both heading towards the engine block.Collaboration: Human and robot work are at the same work piece on different workings tasks. At this stage, collisions are rather likely to occur, and thus the robot must be operated with decreased velocities.Cooperation: After finishing all working tasks at the engine block, the operator carries out other tasks in the common workspace. At this stage, collisions between human and robot are rather unlikely.

### 6.2. Speed Adjustment

Depending on the current separation distance between robot and human, the framework adapts the robot velocity whenever the algorithm-specific separation distance falls bellow the minimum level allowed. The robot’s cycle time $t_{cycle}$ corresponds to the time required from leaving its initial position, conducting all screwing tasks until it returns to the starting position. As an example, the robot movement and the adaptation of robot velocities in dependence of the separation distance while human body part segmentation within the HRSF is being applied are shown in Figure 9a–c. From the plots it can be concluded that the robot is moving with maximum speed during coexistence. When changing to the collaborative phase, reduced velocities are adopted. When the robot returns to its starting position (cooperative phase), the robot velocity can be increased again since the separation distance exceeds the minimum separation distance value allowed.

The cycle time was measured on the robot controller and analyzed for each of the investigated deep learning methods. The obtained results were compared with the cycle times obtained when the (1) state-of-the-art manufacturing safety technology (SICK S30A laser scanner, SICK AG, Waldkirch, Germany) was used or (2) when no additional safety system was applied. In case 1 when the laser scanner was used, the maximum robot velocity, i.e., 1.6 m/s, was reduced to the most restrictive speed value in Table 5. This particular velocity was adjusted constantly for the measurements that dispensed with any additional safety system for human recognition. When no additional safety measure was used (case 2), the lowest speed was applied.

### 6.3. Cycle Time Analysis

For the determination of the algorithm-specific robot process execution times, cycle time measurements of the proposed screwing tasks were repeated ten times for each method investigated. The results obtained are given in Table 6. They clearly indicate that with any of the investigated human body (part) detection methods, the cycle times are always reduced compared to current state-of-the-art implementations of robot collaboration in industrial environments. For example, the cycle time was decreased by up to 35% compared to systems where no additional safety equipment was used. Even when laser scanning systems were used for robot velocity adaptation, the best performing method from the framework achieved cycle time reductions of more than 15%. This is so because laser scanner systems give a very conservative estimate of the intrusion distance *C* and thus a relatively large cycle time.

Among the different algorithms analyzed, both human-body-part-related approaches achieved the lowest levels of robot process execution time. For human pose estimation and human body recognition, higher cycle time fluctuations were observed compared to both segmentation approaches. This behavior can be explained bu both methods reacting less robustly when parts of the human body were occluded, and thus the robot velocity was reduced.

The obtained results clearly show that the HRSF is a valuable approach to reducing process execution times in human–robot collaboration applications. Despite its high latency (compare Table 3), the method of human body part segmentation achieved the lowest mean cycle time results for process execution. Thus, it can be assumed that in the future, the cycle time results can be significantly reduced due to increasing progress in computational power, along with ongoing optimizations in the field of object segmentation.

## 7. Conclusions

This paper describes a performance and feasibility study for vision-based SSM under ISO/TS 15066 constraints. To this end, four different deep learning algorithms were investigated in order to extract spatial human body and human body part information on the basis of RGB-D input data and further to determine separation distances between humans and robots. The regulatory guidelines of ISO/TS 15066 were taken into account, and the algorithm-specific minimum separation distance was used to adapt the robot velocity accordingly.

The framework was applied to screwing tasks and was shown to achieve significant reductions of cycle time compared to state-of-the-art industrial technologies applied for human–robot collaboration today. The best performing method for human body part extraction achieved an optimization of the robot process execution time of more than 15%.

In conclusion, the HRSF provides a promising approach for improving both efficiency and safety in human–robot collaboration. However, before deployment in productive industrial environments, further investigations are required on robustness, reliability, multi-camera integration for plausibility verification, fail-safe scenarios, and functional safety certification. It should be noted that the experimental design has important limitations, including the use of a single subject, a single task, a limited number of repetitions, no formal statistical testing, and simplified baseline comparisons, which constrain the generalization of these initial findings. Addressing these aspects will ensure that the framework is both practically applicable and compliant with industrial safety standards.

## Figures and Tables

**Figure 1 sensors-25-07136-f001:**
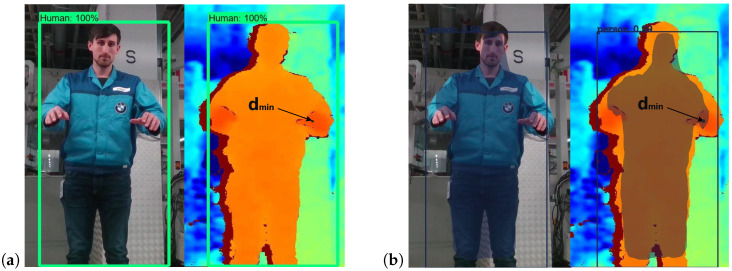
Visualized results of the applied deep learning techniques for (**a**) human body recognition [54] and (**b**) human body segmentation [42] in color image (**left**) and depth image (**right**).

**Figure 2 sensors-25-07136-f002:**
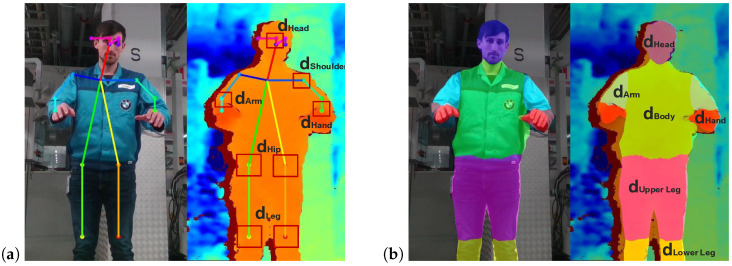
Visualized results of the applied deep learning techniques for (**a**) human pose estimation [45] and (**b**) human body part segmentation [51] in color image (**left**) and depth image (**right**).

**Figure 3 sensors-25-07136-f003:**
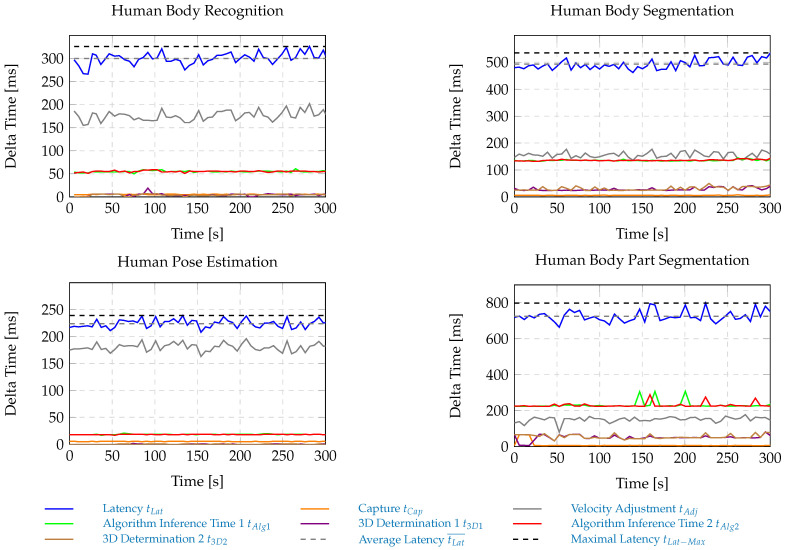
Individual latency contributions determined for each of the analyzed algorithms within the HRSF.

**Figure 4 sensors-25-07136-f004:**
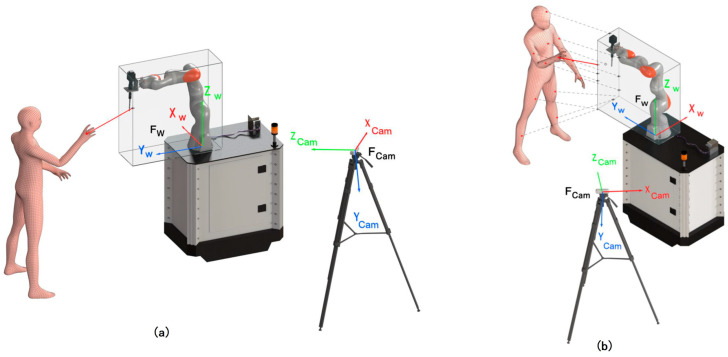
Minimal separation distance determination according to the current robot pose via the application of the cuboid-shaped robot protective hull approach for (**a**) human-body-related and (**b**) human-body-part-related approaches.

**Figure 5 sensors-25-07136-f005:**
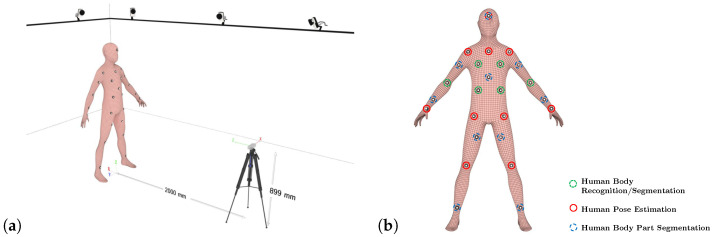
(**a**) Experimental setup for data acquisition. (**b**) Position of markers and indications of whether a marker is used for human pose estimation, human body part segmentation, or only for human-body-related accuracy validation. All of the markers chosen for human pose estimation and human body part segmentation were also considered for human-body-related measurements.

**Figure 6 sensors-25-07136-f006:**
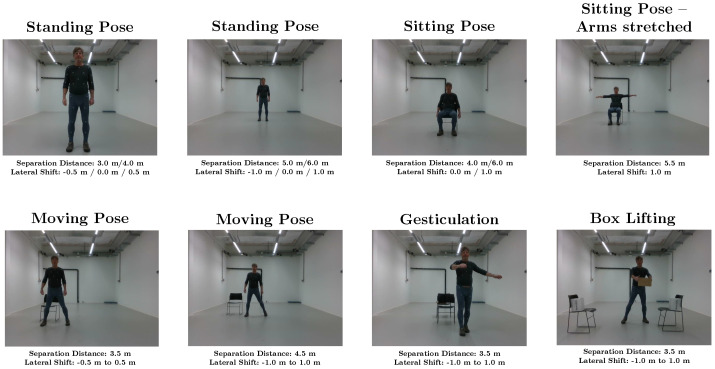
Overview of the different measurement scenarios.

**Figure 7 sensors-25-07136-f007:**
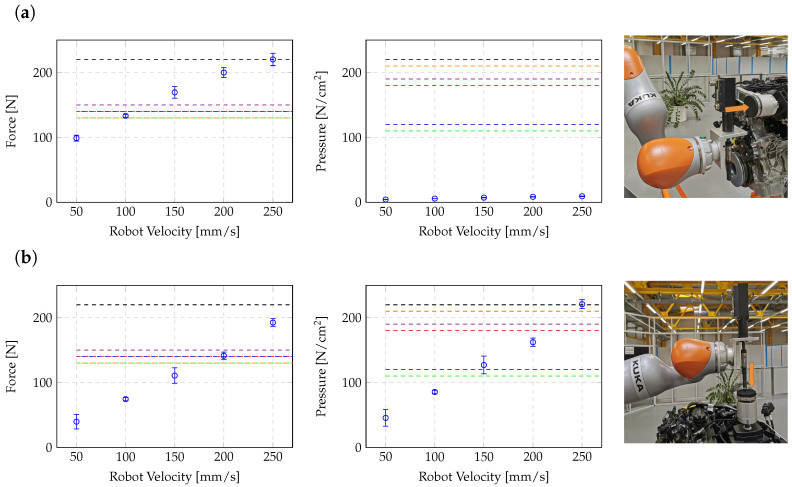
Determination of measured forces and pressures for two robot movement scenarios: movements in operator direction (**a**) and vertical movements (**b**). The colored dashed lines correspond to the biomechanical threshold levels taken from ISO/TS 15066 with the following nomenclature: skull (green), hands/fingers/lower arms (red), chest (blue), upper arms (violet), upper legs (black), and lower legs (orange).

**Figure 8 sensors-25-07136-f008:**
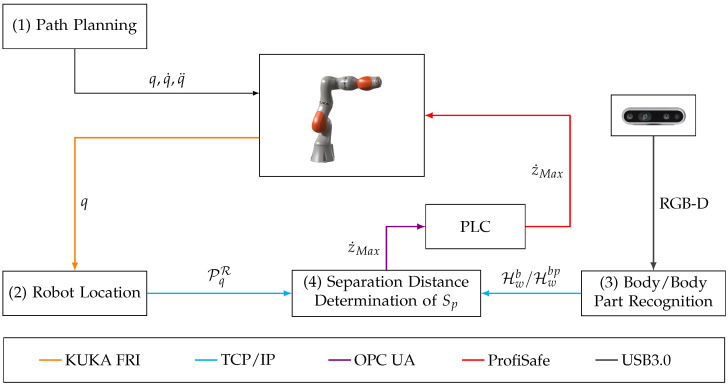
Block architecture of the HRSF using the KUKA iiwa robot.

**Figure 9 sensors-25-07136-f009:**
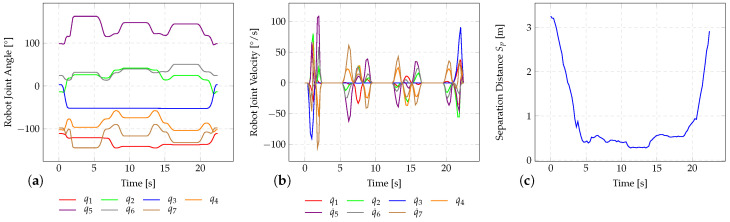
Plotted time evolution of (**a**) robot joint angles, (**b**) robot joint velocities. and (**c**) separation distance $S_{p}$ between the human and robot for the investigated screwing application while the human body part segmentation within the HRSF is applied.

**Table 1 sensors-25-07136-t001:** Comparison of representative vision-based and sensor-based HRC safety approaches.

Work/Approach	Modality	Standard	Body Part	Dynamic Velocity	Experiment
	Addressed	Awareness	Adaptation	Validation	
Industrial safety scanners/light curtains	2D laser/IR	ISO 13855	No	No (fixed stop/speed)	Industrial use; no body part tests
RGB-D human detection [8,9,10]	RGB-D	None or partial	Whole-body only	Limited; coarse scaling	Laboratory demonstrations; limited safety analysis
Skeleton/keypoint tracking [11,12]	RGB/RGB-D	Not aligned with ISO/TS 15066	Joint-level but not mapped to limits	Rarely implemented	Laboratory tests only
DL-based human segmentation for HRC [13,14,15]	RGB-D	Partial references	Region level	Limited/no modulation	Laboratory studies; no latency/failure modes
Depth-based SSM [16,17,18]	Depth, 3D sensors	Partially aligned w/ ISO/TS 15066	Whole-body/coarse regions	Yes, but uniform margins	Strong validation; no body part integration
**Proposed HRSF (this work)**	**RGB-D + DL**	**Explicit ISO/TS 15066 mapping**	**Yes, per-body-part 3D localization**	**Yes, part-specific velocity scaling**	**Real-robot evaluation; accuracy, latency, robustness**

**Table 2 sensors-25-07136-t002:** Analyzed deep learning approaches for human body and human body part recognition.

Detection Algorithm	Method
Human Body Recognition	SSD [54]
Human Body Segmentation	Mask R-CNN [42]
Human Pose Estimation	Deep Pose [45]
Human Body Part Segmentation	Human Body Part Parsing [51]

**Table 3 sensors-25-07136-t003:** Contributions of the analyzed algorithms to the separation distance due to the maximal latency within the HRSF.

Method	$t_{\mathrm{Lat}-\mathrm{Max}}$	$S_{h}$
	[ms]	[mm]
Human Body Recognition	370	592
Human Pose Estimation	305	488
Human Body Segmentation	559	894
Human Body Part Segmentation	812	1299

**Table 4 sensors-25-07136-t004:** Determined position prediction error for all algorithms applied in the HRSF.

Method	Δx	Δy	Δz
	[mm]	[mm]	[mm]
Human Body Recognition	346	767	399
Human Body Segmentation	346	416	334
Human Pose Estimation	131	57	206
Human Body Part Segmentation	87	71	151

**Table 5 sensors-25-07136-t005:** Body-part-specific maximum Cartesian robot velocities that were applied in the HRSF for the analyzed screwing application.

Body Part	${\dot{z}}_{\mathrm{Max}}$ [mm/s]
Skull/Forehead	50
Hand/Fingers/Lower Arms	100
Chest	100
Upper Arms	100
Thighs	200
Lower Legs	50

**Table 6 sensors-25-07136-t006:** Overview of robot cycle times obtained for the execution of screwing tasks with different algorithms applied in the HRSF, state-of-the-art laser scanners, and no additional safety monitoring equipment being used.

Method	$t_{\mathrm{Cycle}}$ [s]
Human Body Recognition	24.84 ± 2.31
Human Body Segmentation	25.60 ± 0.41
Human Pose Estimation	22.81 ± 1.17
Human Body Part Segmentation	22.78 ± 0.37
Laser Scanner	26.98 ± 0.59
No Additional Safety System	35.08 ± 0.04

## Data Availability

The original contributions presented in this study are included in the article. Further inquiries can be directed to the corresponding author.

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
