# Peer review of "Analysis of Deep-Learning Methods in an ISO/TS 15066–Compliant Human–Robot Safety Framework"

_sensors, 2025, doi:10.3390/s25237136_

Round 1
Reviewer 1 Report
Comments and Suggestions for Authors
Abstract
-
Lack of clarity on novelty: The abstract merely summarizes the framework and tested algorithms but does not explicitly state what is new compared to existing ISO/TS 15066-compliant studies.
-
Absence of quantitative context: The claim of “significant reduction in process execution time” lacks numerical support or statistical validation.
-
No mention of limitations: The abstract reads as purely positive, omitting discussion of computational latency, scalability, or real-world validation concerns.
-
1. Introduction
-
Superficial problem framing: While it references ISO/TS 15066, it does not rigorously justify why deep learning offers a superior solution compared to traditional sensor-based methods.
-
Weak research gap articulation: The introduction lists prior works but fails to critically identify a precise methodological or empirical gap the proposed HRSF addresses.
-
No clear hypothesis: The study lacks a guiding research question or hypothesis linking deep learning methods to measurable safety improvements.
-
Overreliance on general statements: Phrases like “it is important to distinguish humans from other moving objects” are obvious and add little analytical value.
-
2. Safety Aspects According to ISO/TS 15066
-
Limited theoretical depth: The section paraphrases ISO/TS 15066 without critically engaging its practical limitations or compliance challenges.
-
Over-simplification of safety logic: The assumption that dynamic speed adaptation alone ensures compliance is unconvincing, given ISO/TS 15066’s strict biomechanical testing requirements.
-
No discussion of uncertainty or certification issues: The study does not consider how uncertified AI models could compromise regulatory validation.
-
3. Localization of Humans and Human Body Parts in the Workspace
-
Fragmented structure: The subsections are overly descriptive, summarizing algorithms rather than critically evaluating their suitability for real-time industrial contexts.
-
Absence of selection rationale: The chosen algorithms (SSD, Mask R-CNN, DeepPose, Human Body Part Parsing) are presented without justification for why these specific architectures were selected over more recent or lightweight models.
-
No performance benchmarks: The section lacks comparative metrics (e.g., inference speed, accuracy trade-offs) to substantiate claims of effectiveness.
-
Insufficient treatment of dataset bias: No discussion of domain adaptation challenges when transferring models trained on MS COCO or PASCAL-Part to manufacturing environments.
-
Unrealistic assumptions: The study assumes controlled lighting and limited occlusion, which are rarely representative of industrial shop floors.
-
4. Determination of ISO-Relevant Safety Parameters
-
Overly formulaic and under-validated: Equations from ISO standards are reproduced without demonstrating empirical verification of parameter estimation.
-
Inadequate error analysis: While latency and uncertainty are mentioned, there is no statistical treatment (confidence intervals, standard deviations, or sensitivity analysis).
-
Lack of scalability consideration: Safety parameter tuning is highly specific to one robot (KUKA iiwa), making the framework’s generalizability questionable.
-
Experimental design weakness: The latency tests lack clarity on repetition count, variance, or environmental conditions influencing measurement.
-
Neglected human variability: Human motion speeds are fixed to standard ISO assumptions, ignoring real inter-person variability that affects safety margins.
-
5. The Human-Robot-Safety Framework (HRSF)
-
Conceptual rigidity: The block architecture is hardware-locked to KUKA iiwa, undermining claims of general applicability.
-
No real-time validation: The communication between modules (e.g., latency over OPC UA) is described but not stress-tested under industrial loads.
-
Lack of redundancy mechanisms: The framework has no provision for fail-safe recovery if the deep-learning node fails or misclassifies human presence.
-
No security considerations: Data transfer between modules via TCP/IP and OPC UA introduces potential safety/security vulnerabilities not discussed.
-
No functional safety certification path: The study ignores ISO 13849 or IEC 61508 standards that govern safety-rated software systems.
-
6. Experimental Validation
-
Narrow test case: The “screwing task” represents a trivial scenario and does not capture diverse collaborative operations or multiple human operators.
-
Small sample size: Only one test environment and presumably one participant are used; hence, results lack statistical or ecological validity.
-
No ground-truth validation for predictions: There is no report on how predicted human positions were verified beyond motion capture, nor on occlusion robustness.
-
Lack of computational benchmarking: Claims about “latency reduction” and “real-time adaptation” are unsubstantiated by FPS rates or computational resource profiling.
-
Cycle time focus too narrow: Process efficiency is emphasized over safety integrity, neglecting whether the deep-learning methods could misclassify dangerous situations.
-
No ablation or comparative experiments: There is no analysis of how individual framework components (e.g., segmentation accuracy vs. latency) contribute to overall performance.
-
-
-
-
-
-
Reviewer 2 Report
Comments and Suggestions for Authors
The paper presents a well-structured and technically sound study on a deep-learning-based Human-Robot Safety Framework (HRSF) compliant with ISO/TS 15066. It systematically compares four deep learning approaches—human body recognition, segmentation, pose estimation, and body part segmentation—and evaluates their impact on safety and efficiency in collaborative robotics. The topic is highly relevant to current industrial automation trends and human-robot collaboration research.
- The paper acknowledges the issue "Functional Safety of Deep Learning Models" but does not provide any formal risk assessment or redundancy validation (e.g., fail-safe mechanisms).
- More details are needed on how dataset bias, lighting variation, or occlusion affect recognition accuracy.
- The results section would benefit from statistical significance testing (e.g., ANOVA) to confirm observed improvements.
- Only one industrial baseline (laser scanner) is tested. Additional comparisons with other vision-based safety systems would strengthen the claim.
- Although technically detailed, some sections (especially 3.1–3.3) are dense and could be streamlined for readability.
- The title is too long.
Reviewer 3 Report
Comments and Suggestions for Authors
This research developed and experimentally validated a Human-Robot Safety Framework (HRSF) integrating multiple deep learning-based human body and body-part recognition methods to dynamically adjust robot velocities in compliance with ISO/TS 15066 safety standards. The study systematically compared four state-of-the-art neural models (SSD, Mask R-CNN, DeepPose, and Body Part Parsing) using RGB-D data to optimize safety-aware human–robot collaboration in manufacturing, achieving up to 35% reduction in process cycle time relative to conventional safety systems.
Here are my comments:
The study relies on a single experimental setup with only ten repetitions, which limits the statistical strength of the reported findings.
The dataset selection further constrains generalizability, as training and evaluation were conducted solely on MS COCO and PASCAL-Part. These datasets do not adequately represent the complex visual conditions of industrial settings, where factors such as variable lighting, occlusion, and differing worker appearances play a significant role.
The inference latency, reported between 300 and 800 milliseconds, remains too high for real-time collaboration between humans and robots. Lack of optimization for general-purpose hardware or techniques such as model compression, pruning.
The research does not include formal validation for functional safety according to industrial standards. The absence of demonstrated compliance with key frameworks such as ISO 13849 or IEC 61508 creates uncertainty about the framework’s readiness for certified use in safety-critical manufacturing environments.
Several important cross-disciplinary literatures are missing, like Prediction model of dynamic fracture toughness of nickel-based alloys; and Funabot-Sleeve: A Wearable Device Employing McKibben Artificial Muscles for Haptic Sensation in the Forearm.
Collision modeling is simplified to quasi-static contact scenarios and excludes dynamic or elastic interactions.
The proposed control system architecture, based on OPC UA and PLC communication, is conceptually clear but has not been validated under real-world network delay or fault conditions.
Reproducibility is not good because of the absence of shared implementation details, including algorithmic parameters, calibration procedures, and source code availability.
Providing algorithmic pseudocode or a detailed control-loop description.
Round 2
Reviewer 1 Report
Comments and Suggestions for Authors
Thanks for the revisions and glad to see the improvement. However, novelty and positioning are not sufficiently differentiated from existing vision-based HRC safety frameworks and HRC path-planning literature. Functional safety and standards integration are discussed conceptually but not demonstrated rigorously; In my previous review, I emphasized this needs to be well synthesized.
Also, the Experimental design has important limitations (single subject, single task, limited repetitions, no statistical testing, simplified baselines. Some that cant be addressed can be included in the limitations of the study. I also raised the issue of depth and robustness analysis are incomplete relative to what would be needed for safety-critical deployment (false negatives, worst-case scenarios, multi-human situations). The research gap remains somewhat generic. Vision-based safety systems for HRC using RGB-D and deep learning are already well studied; you cite several relevant works but do not clearly articulate what those methods cannot do that your framework does (beyond “body-part aware” velocity limits).
The benefits attributed to body-part–specific segmentation vs whole-body detection are plausible, but the paper does not provide clear analytical or empirical evidence that body-part granularity is the primary driver of cycle-time improvements, especially given the large latencies of those methods. Add a “Contributions” subsection in the Introduction that clearly lists 3–4 specific, novel contributions and explicitly contrasts them with the closest existing work on vision-based SSM and deep-learning HRC safety.. Then, onsider including a comparative table of related work (sensing modality, standard addressed, body-part awareness, dynamic velocity adaptation, experimental validation) and highlight where your HRSF is different
Clarify the status of the framework: “conceptual safety architecture and performance analysis” vs “certified, deployable safety system.” At present, it is clearly the former, and that should be stated explicitly. Reframe the work as a performance and feasibility study for vision-based SSM under ISO/TS 15066 constraints, without suggesting actual compliance
Please Provide more detail on experimental conditions. The motion-capture-based accuracy evaluation is a strong point, but the treatment can be strengthened
Clarify exactly how Table 3 is computed and separate systematic bias from random noise (offset vs. variance),
Reviewer 3 Report
Comments and Suggestions for Authors
I have reviewed the revised manuscript and I am pleased to report that the authors have made substantial improvements in response to my previous comments. The manuscript now addresses the key concerns I raised, and the revisions enhance both the clarity and depth of the work. I recommend that the manuscript be accepted for publication in its current form.
Author Response
Thank you for the recommendation for publication.